# Microtubules soften due to cross-sectional flattening

Edvin Memet[1†], Feodor Hilitski[2†], Margaret A Morris[2], Walter J Schwenger[2], Zvonimir Dogic[2,3], L Mahadevan[1,4,5]*

[1]Department of Physics, Harvard University, Cambridge, United States; [2]Department of Physics, Brandeis University, Waltham, United States; [3]Department of Physics, University of California, Santa Barbara, Santa Barbara, United States; [4]Paulson School of Engineering and Applied Sciences, Harvard University, Cambridge, United States; [5]Kavli Institute for Nano-Bio Science and Technology, Harvard University, Cambridge, United States

**Abstract** We use optical trapping to continuously bend an isolated microtubule while simultaneously measuring the applied force and the resulting filament strain, thus allowing us to determine its elastic properties over a wide range of applied strains. We find that, while in the low-strain regime, microtubules may be quantitatively described in terms of the classical Euler-Bernoulli elastic filament, above a critical strain they deviate from this simple elastic model, showing a softening response with increasingdeformations. A three-dimensional thin-shell model, in which the increased mechanical compliance is caused by flattening and eventual buckling of the filament cross-section, captures this softening effect in the high strain regime and yields quantitative values of the effective mechanical properties of microtubules. Our results demonstrate that properties of microtubules are highly dependent on the magnitude of the applied strain and offer a new interpretation for the large variety in microtubule mechanical data measured by different methods.
DOI: https://doi.org/10.7554/eLife.34695.001

*For correspondence:
lmahadev@g.harvard.edu

[†]These authors contributed equally to this work

**Competing interests:** The authors declare that no competing interests exist.

## Introduction

Microtubules (MTs) are long slender hollow cylindrical filaments with an approximate inner and outer diameter of 15 nm and 25 nm (*Nogales et al., 1999*). They are an indispensable structural element in biology, and their mechanical properties play an critical role in defining the shape and functionalities of various cellular architectures including neuronal axons, cilia and flagella, centrioles as well as the mitotic spindle (*Howard, 2001*). Therefore, a quantitative understanding of their mechanical properties is essential for elucidating the properties of various biological structures and functions (*Schaedel et al., 2015*; *Wells and Aksimentiev, 2010*; *Schaap et al., 2006*). A number of experimental studies have measured either the flexural rigidity ($EI$) or persistence length ($l_P$) of MTs, two closely related quantities that determine a filament's resistance to bending (see *Hawkins et al., 2010*). However, there is a considerable disagreement between different reported values of MT elastic moduli (*Mickey and Howard, 1995*), the dependence of the elastic moduli on the presence of stabilizing agents (taxol and GMPCPP), as well as a possible length-dependence of flexural rigidity (*Pampaloni et al., 2006*; *Kis et al., 2002*).

In principle, the properties of microtubules can be measured by either visualizing their intrinsic thermal fluctuations (*Gittes et al., 1993*; *Mickey and Howard, 1995*; *Pampaloni et al., 2006*; *Brangwynne et al., 2007*; *Janson and Dogterom, 2004*; *Cassimeris et al., 2001*) or by applying an external force through optical trapping or hydrodynamic flow experiments (*Kikumoto et al., 2006*; *van Mameren et al., 2009*; *Van den Heuvel et al., 2008*; *Venier et al., 1994*; *Felgner et al., 1996*; *Kurachi et al., 1995*; *Dye et al., 1993*). Because of significant rigidity of MTs these two types of

measurements typically probe different deformation (strain) regimes. Thermally induced fluctuations only induce small strain deformations, while methods that use external forces are more suited to probe the high-strain regime. The existing measurements of microtubule mechanics have been extensively reviewed iteHawkins2010. Careful review of this data shows that the majority of measurements that rely on the filament fluctuations in the low strain regime yield values of microtubule flexural rigidity around , while measurements at high strains, tend to be for similarly prepared MTs (*Supplementary file 2*).

Here we describe experiments that comprehensively probe the mechanical response of GMPCPP stabilized microtubules across a wide range of imposed strains. Using optical trapping, we attach micron-sized silica beads at different points along a single filament and subject it to tensile and compressive forces. This allows us to construct a force-strain relationship for an individual filament that connects the low and high strain regimes that were probed separately in previous experiments. In the compression region, which corresponds to microtubule bending, the buckling force increases linearly in the initial (low strain) region, but quickly deviates from this trend and saturates at some critical strain, and it gradually decreases at high strain values. Such behavior indicates softening of MTs, an outcome that cannot be captured by the ideal Euler-Bernoulli model. We show that the mechanical response of MTs is well-captured by an anisotropic elastic shell model with three coarse grained elastic parameters (oneshear, and two stretching moduli). Our numerical simulations demonstrate that the softening in the high strain regime can be ascribed to cross-sectional ovalization and eventual buckling, an effect first described for macroscopic hollow cylinders by Brazier (*Brazier, 1927*; *Calladine, 1983*). For microtubule filaments in particular, cross-sectional flattening and buckling in response to compressive radial strains has also been described (*Kononova et al., 2014*).

To test the validity of our method we calibrate our experiments and theory using a different biological filament - the bacterial flagellum. Similar to microtubules, flagella are long-filamentous cylindrical biopolymers with an outer diameter of $\sim 24$ nm (*Yonekura et al., 2003*). Furthermore, like microtubules flagella have a hollow core; however, their core diameter is only 2 nm. Therefore flagella should effectively behave as solid cylinders. Most wild-type flagella have a characteristic helical superstructure. We use flagellin monomers isolated from Salmonella typhimurium, strain SJW1660, which have a point mutation in their amino-acid sequence that causes them to assemble into straight filaments (*Kamiya et al., 1980*). A simple model based on Euler-Bernoullibeams quantitatively describes the measured force curve in all strain regimes. Such measurements validate our experimental method while also providing an estimate of the elastic properties of flagellar filaments that are in reasonable agreement with thefew existing measurements (*Louzon et al., 2017*; *Darnton and Berg, 2007*; *Fujime et al., 1972*).

## Results

### Optical tweezers buckle microtubules

Optical trapping techniques have been used to manipulate and characterize diverse biological systems, including individual biological filaments and their assemblages (*van Mameren et al., 2009*; *Hilitski et al., 2015*; *Wang et al., 1997*; *Darnton and Berg, 2007*). We use independently controlled optical traps to attach a pair of neutravidin-coated silica beads ($D = 1\,\mu\mathrm{m}$) to a single, segmented partially biotinylated MT filament (*Figure 1A*). One of the traps - *the detection trap* - is held stationary and used as a force detector by measuring the displacement of its captured bead with back focal- plane interferometry (see *Video 1* of the experiments in supplementary information). The second optical trap - *the manipulation trap* - is moved in increments of 2–20 nm in order to exert a force on the filament, which connects the two beads. Our instrumentation allows us to simultaneously apply an external force with optical tweezers and visualize the filament configuration (*Figure 1B*). Strain $\epsilon$ is defined as:

$$\epsilon = (d_b - L)/L. \qquad (1)$$

where $d_b$ is the bead separation once the MT has been deformed, and $L$ is the distance between the beads when there is no force on the MT. The measured force-strain curve exhibits an unexpected feature (*Figure 2B*). For small deformations, the force increases linearly as a function of the applied strain. Surprisingly, above a critical applied strain ($\epsilon \sim -0.1$) the force-compression curve

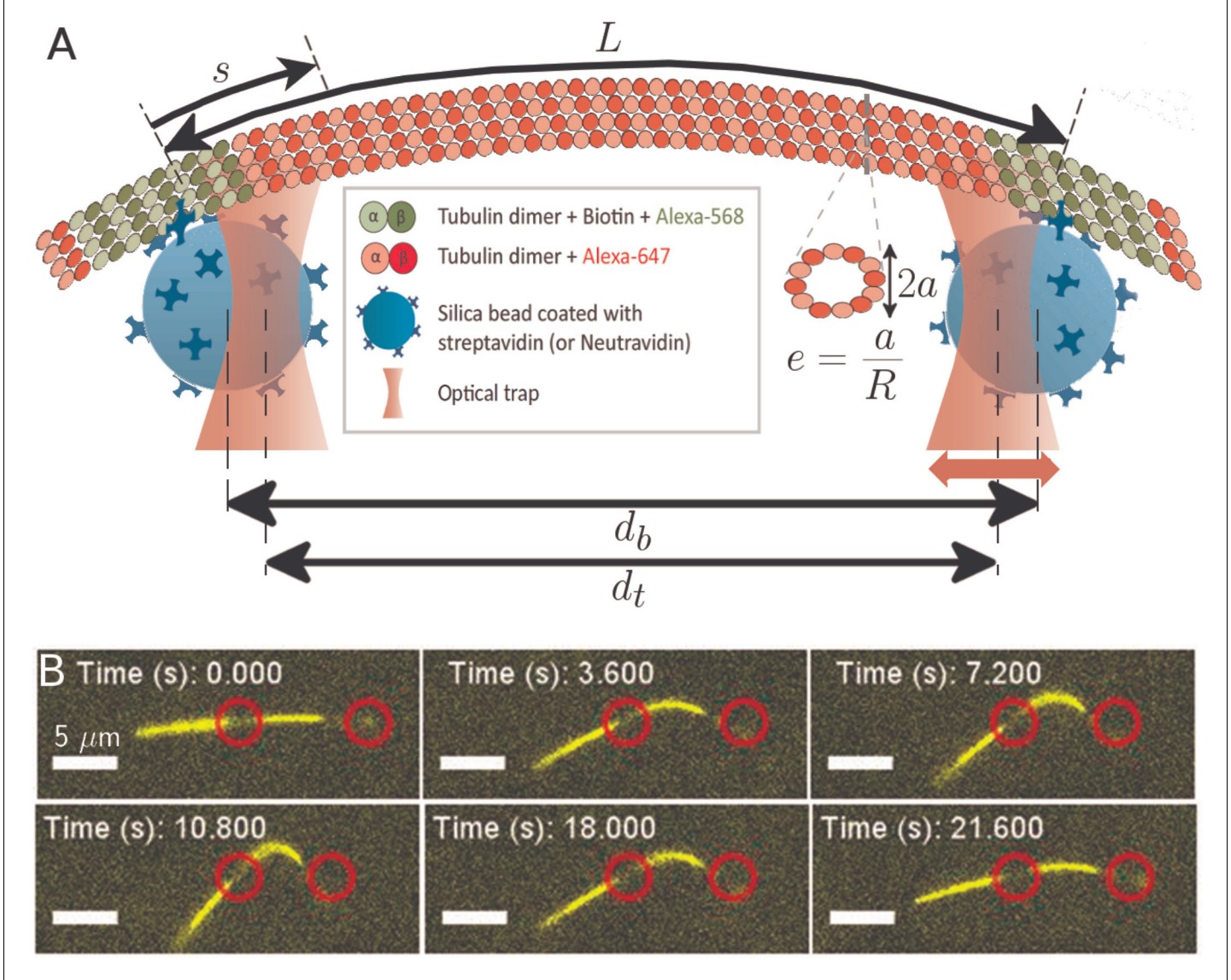

**Figure 1.** Probing the bending and stretching of microtubules using an optical trap. (A) Schematic of the experimental setup for microtubule compression and stretching (note: not to-scale). Beads of radius 0.5 $\mu$m manipulated through optical traps are attached to the filament via biotin-streptavidin bonds. The left trap remains fixed, while the right one is incrementally displaced towards or away. The bead separation $d_b$ is different from the trap separation $d_t$ due to the displacement between the centers of each bead and the corresponding trap. The force exerted by the traps on the beads is proportional to this displacement, with the constant of proportionality being the optical trap stiffness. The (effective) MT length $L$ is the length between the attachment points of the two beads (but note that multiple bonds may form). The arclength coordinate $s$ is measured from the left attachment point and takes value between 0 and $L$: $0<s<L$. A cross-sectional profile is shown to illustrate our definition of 'eccentricity' or 'flatness' $e = (R - a)/R$ (**Equation 4**) where $2a$ is the length of the small axis and $R$ is the radius of the undeformed MT cross-section, taken to be $R = 12$ nm in our simulations. (B) Series of fluorescence images from microtubule compression experiment. Trap positions represented through red circles of arbitrary size are drawn for illustration purposes and may not precisely represent actual trap positions, since neither beads nor traps are visible in fluorescence imaging. The mobile trap is moved progressively closer to the detection trap in the first four snapshots, resulting in increasingly larger MT buckling amplitudes. The mobile trap then reverses direction, decreasing the buckling amplitude as seen in the last two snapshots. The scale bar represents 5 $\mu$m, the end-to-end length of the MT is approximately 18 $\mu$m, and the length of the MT segment between the optical beads is 7.7 $\mu$m.
DOI: https://doi.org/10.7554/eLife.34695.002

exhibits a qualitatively different behavior. In this regime the applied force remains flat or in some cases it even declines slightly, even as the strain changes from −0.1 to −0.5. Filaments subjected to repeated cycles of compression/extension do not seem to exhibit any aging (**Figure 2B**, inset).

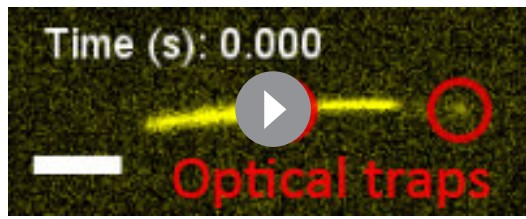

**Video 1.** Microtubule buckling experiment, showing two 'back-and-forth' cycles. The microtubule is labeled with fluorescent dye Alexa-647 and is actively kept in the focal plane. The scale bar represents 5 $\mu$m. Approximate positions of the optical traps are shown by red circles of arbitrary size.
DOI: https://doi.org/10.7554/eLife.34695.003

## Flagellar filaments exhibit no anomalous softening at high strains

The softening of microtubules at high-strains is not predicted by simple elastic filament models of microtubule elasticity. A plausible hypothesis is that the hollow core of the microtubules significantly changes the effective elastic properties of the filament. To examine this hypothesis and quantitatively test our experimental method we measured the force-strain curve of bacterial flagellar filaments. As mentioned previously, the hollow core of flagellar filaments is only 2 nm. Therefore, we expect theywill exhibit classical slender-rod-like behavior that should be captured by a one-dimensional filament model.

The measured strain-force curve of a flagellar filament is qualitatively different from that observed for microtubules (*Figure 3B*). In both the compression and extension regime, the force monotonically increases. For classical buckling, the amplitude of thedeformed beam scales as $\sqrt{F - F_c}$, where $F_c = \pi^2 B/L^2$ is the classical buckling force (*Timoshenko and Gere, 2012*). For small deflections, since the strain goes like the square of the amplitude, the force-displacement curve should be roughly linear with strain, starting at $F_c$ (albeit growing very slowly). However, for the classical buckling analysis, the compressive force is assumed to apply directly along the centerline of the beam. Experimentally, this is not the case, since the force is exerted on the filament through trapped micron-sized beads, which displace the force contact points away from the beam's long axis. Therefore to quantitatively compare experiments to theory we numerically solved a more realistic problem that accounts for the entire filament-bead configuration. We also note that the slope in the extension region is higher than that in the compression region. Because the force is not applied along the filament centerline, in response to extensional forces the filament bends in the proximity of the bead attachments point. It is more difficult to bend the filament into such a configuration, hence the larger slope of the extensional regime.

By tuning a single parameter, flexural rigidity $B$, the force-strain curve obtained from simulations of a 1D filament model can be quantitatively fitted to the experimental data (*Figure 3B*). In the compressive regime, the data is well fitted by simulationsin which the bending rigidity $B = 4\text{pN} \times \mu m^2$. Note that the agreement extends to very large strains approaching $-0.7$. The fitting procedure was repeated for eight distinct flagella filaments, yielding flexural rigidity that ranges from 3.2 to 5.25 $\text{pN} \times \mu m^2$, with a mean around $4.1 \pm 0.6 \, \text{pN} \times \mu m^2$,. Interestingly, this value is close to the value of

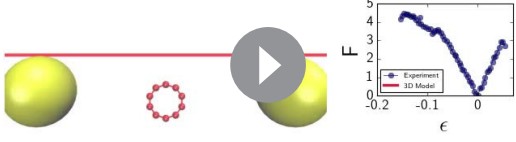

**Video 2.** Flagellum buckling experiment. (left) Simulation of microtubule buckling experiment shown in Figure 4, first panel. The length of the microtubule between the optical bead attachment points is 3.6 $\mu$m while the size of the optical beads is 0.5 $mu$m. The shape of the cross-section located halfway along the microtubule is also shown. (right) Force-strain curves from experiment (*blue*) and simulation (*red*). The simulation curve is synchronized with the visualization on the left.
DOI: https://doi.org/10.7554/eLife.34695.004

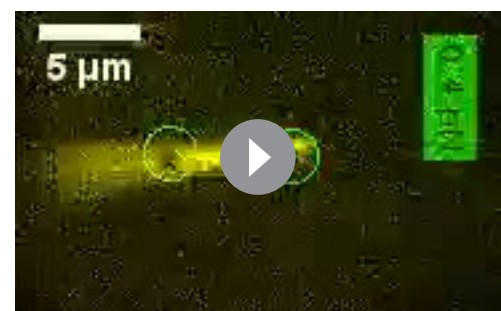

**Video 3.** Flagellum buckling experiment, showing three 'back-and-forth' cycles. The scale bar represents 5 $\mu$m. Approximate positions of the optical traps are shown by green circles of arbitrary size. Measured force vector and magnitude are also indicated.
DOI: https://doi.org/10.7554/eLife.34695.005

$3.5\text{pN} \times \mu m^2$ and $2.9\text{pN} \times \mu m^2$ reported, respectively for the coiled and straight forms of flagella (*Louzon et al., 2017*; *Darnton and Berg, 2007*). The magnitude of $B$ suggests that Young's modulus for the straight flagellum is GPa.

## 3d model explains microtubule softening

Quantitative agreement between experimentally measured force-strain curves of flagellar filaments and simulation results validate our experimental technique, while also demonstrating that the flagellar filaments behave as simple elastic filaments. However, the same one-dimensional model is not capable of reproducing the more complex elastic behavior observed in microtubule filaments. At most, it can describe the low strain regime of the force-compression curve (*Figure 2B*, green dashed curve). To explain theelastic behavior of microtubules over the entire strain regime we developed a more comprehensive 3D model of microtubules that explicitly accounts for their hollow center. We model the microtubule as a 3D orthogonal network of springs wrapped into a cylinder (*Figure 2A*). All interactions between particles are defined in terms of a stretching energy $V_{\text{stretch}}(k, l, l_0)$ or a bending energy $V_{\text{bend}}(\kappa, \phi, \phi_0)$, where $l$ is the distance between a particle pair, $\phi$ is the angle defined by a triplet, and the subscript' zero' denotes the separation that minimizes particle interaction energy. Explicitly,

$$V_{\text{stretch}}(k, l, l_0) = \frac{1}{2}k(l - l_0)^2, \tag{2}$$

$$V_{\text{bend}}(\kappa, \phi, \phi_0) = \frac{1}{2}\kappa(\phi - \phi_0)^2. \tag{3}$$

Neighboring triplets in the axial and azimuthal directions encode bending energies (*Table 1*) through the parameters $\kappa_a$ (*Figure 2A*, dark blue), respectively $\kappa_c$ (*Figure 2A*, dark green) and equilibrium angles of $\pi$, respectively $144\pi/180$. Shearing is encoded by the same type of energy function as bending (*Table 1*), but with parameter $\kappa_s$ (*Figure 2A*, brown) and rest angle $\pi/2$: $V_{\text{shear}}(\kappa_s, \phi_s) = V_{\text{bend}}(\kappa_s, \phi_s, \pi/2)$. The shearing interaction is defined for any triplet of neighbors that are not all along the axial or azimuthal directions. The microtubule-optical bead contact is modeled via parameters $k_b$ and $\kappa_b$, which are made sufficiently large to enforce the constraints of inextensibility and fixed attachment point. The relationship between these microscopic parameters and the macroscopic parameters $E_a$, $E_c$, and $G$ (axial Young's modulus, circumferential Young's modulus, and shear modulus) can be derived based on the planar spring-network model and the orthotropic elastic shell model (*Sim and Sept, 2013*; *Wang et al., 2006*) and are shown in *Table 1*.

The 3D microtubule model quantitatively explains the softening that is observed in experimentally measured force-strain curves of microtubules. Overall we investigated the properties of 10 different filaments with lengths between 3 and 15 $\mu$m (*Figure 4*) and fitted 10 different force-strain curves to our theoretical model. Fitting each measurement curve yields independent yet consistent estimates of the Young's modulus in axial and circumferential directions, $E_a$, $E_c$, as well as the shear modulus, $G$. We obtain $E_c$ that varies between 3–10 MPa, and $E_a$ between 0.6–1.1 GPa, which is in reasonable agreement with values reported in the literature (see Discussion). This observation confirms that microtubules are highly anisotropic materials. Interestingly, our values for shear modulus, $G$, are in the GPa range (*Table 2*), which is significantly higher than the largest values reported in the literature. This likely indicates that the microtubule deformations in our experiments are large enough to reach the point of deforming the tubulin units themselves rather than the bonds between them (see Discussion).

## Cross-sections flatten and eventually buckle

With quantitative agreement between experiments and theory we are in a position to elucidate the microscopic origin of the microtubule softening at high strains. The decrease in the buckling force is directly associated with ovalisation of the microtubulecross-section. Plotting the microtubule cross-section at different deformation strains effectively demonstrates this effect. *Figure 5A* show the force-strain data for a microtubule of length 8.3 $\mu$m. The point labeled *a* is close to the boundary of the region where the 1D model (*Figure 5A*, green dashed line) fails, while points *b* and *c* are both inside the high-strain non-classical regime. The large inset shows a snapshot of the

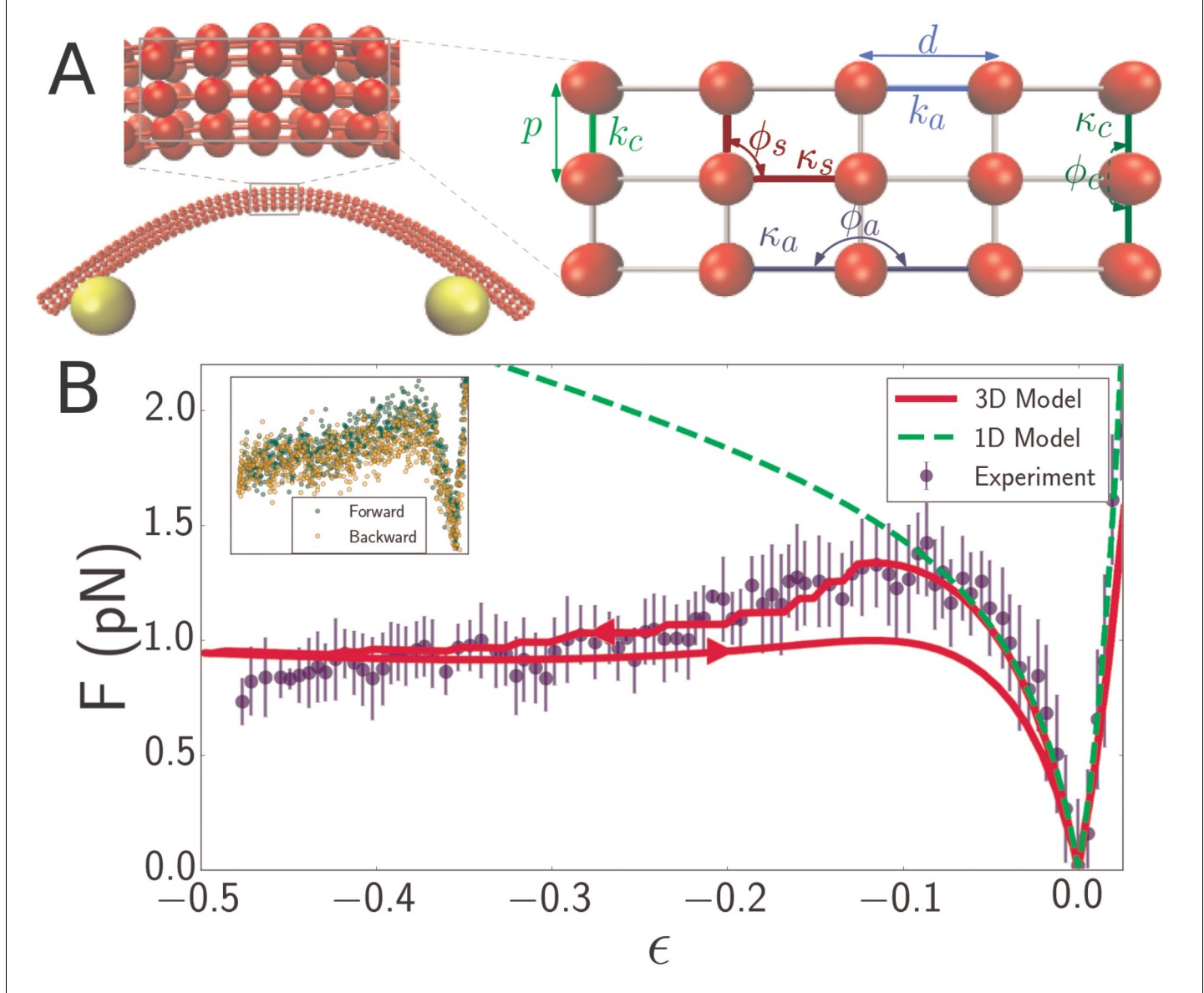

**Figure 2.** Discrete mechanical network model of microtubule. (**A**) Schematic of spring network model of microtubule. Red particles discretize the filament while yellow particles represent the optical beads. Top inset: closeup of a region of the microtubule model, showing the 3D arrangement of the particles. Right inset: further closeup of the region, showing the defined interactions between particles. In the circumferential direction, ten particles are arranged in a circle of radius $R = 12\,\text{nm}$, (only three particles are shown in the inset) with neighboring pairs connected by springs of stiffness $k_c$ and rest length $p$ (light green). Neighboring triplets in the circumferential direction form an angle $\phi_c$ and are characterized by a bending energy interaction with parameter $\kappa_c$ (dark green). In the axial direction, cross-sections are connected by springs of stiffness $k_a$ (light blue) running between pairs of corresponding particles. Bending in the axial direction is characterized by a parameter $\kappa_a$ and the angle $\phi_a$ formed by neighboring triplets (dark blue). All other triplets of connected particles that do not run exclusively in the axial or circumferential direction encode a shearing interaction with energy given by $V_{\text{bend}}(\kappa_s, \phi_s, \pi/2)$ (**Table 1**) with parameters $\kappa_s$ and $\phi_s$ (yellow), where the stress-free shear angle is $\pi/2$. (**B**) Force-strain curve for a microtubule of length 7.1 $\mu m$ from experiment (blue) and simulation using a 1D model (green, dashed) or a 3D model (red). Arrows on the red curve indicate forward or backward directions in the compressive regime. The bending rigidity for the 1D model fit is $B = 12\text{pN}\mu\text{m}^2$. The fit parameters for the 3D model are $E_a = 0.6$ GPa, $E_c = 3$ MPa, and $G = 1.5$ GPa. Errorbars are obtained by binning data spatially as well as averaging over multiple runs. Inset: Raw data from individual 'forward' (green) and 'backward' (yellow) runs.

DOI: https://doi.org/10.7554/eLife.34695.006

simulatedmicrotubule corresponding to point **c** and smaller insets show cross-sectional profiles at

**Table 1.** List of microscopic parameters for the 3D model, their physical significance, the energy functional associated with them (**Equation2** or **Equation 3**), and their connection to the relevant macroscopic elastic parameter.

Parameters, angles, and lengths are highlighted in **Figure 2A**. where $h \approx 2.7$ nm and $h_0 \approx 1.6$ nm are the microtubule's thickness and effective thickness (see Materials and methods), respectively and factors involving the Poisson ratios $\nu_a$ and $\nu_c$ are neglected.

| Microscopic Parameter | Physical Significance | Associated Energy | Relation to Macroscopic moduli |
|---|---|---|---|
| $k_a$ | Axial stretching | $\frac{1}{2}ka(l_a - d)^2$ | $\frac{E_a h p}{d}$ |
| $\kappa_a$ | Axial bending | $\frac{1}{2}\kappa_a(\phi_a - \pi)^2$ | $\frac{1}{6}\frac{E_a h_0^3 p}{d}$ |
| $k_c$ | Circumferential stretching | $\frac{1}{2}k_c(l_c - p)^2$ | $\frac{E_c h d}{p}$ |
| $\kappa_c$ | Circumferential bending | $\frac{1}{2}\kappa_c(\phi_c - 144\pi/180)^2$ | $\frac{1}{6}E_c h_0^3 d/p$ |
| $\kappa_s$ | Shearing | $\frac{1}{2}\kappa_s(\phi_s - \pi/2)^2$ | $2G\,hpd$ |

DOI: https://doi.org/10.7554/eLife.34695.008

indicated locations. While cross-sectional deformations vanish at the endpoints (i.e. at the locations of the optical beads, shown in blue), deformations in the interior are considerable.

To quantify the cross-sectional deformation profiles more explicitly, we define a measure of cross-sectional eccentricity as:

$$e = \frac{R - a}{R}, \tag{4}$$

where $a$ is semi-minor axis (**Figure 1A**) and $R = 12$ nm is the radius of the undeformed microtubule cross-section. $e = 0$ corresponds to a circular cross-section, while larger values of $e$ correspond to increasingly flatter cross-sections. The cross-sectional profile of a buckled filament in the low-strain regime (point $a$ in **Figure 5A**) has $e \approx 0$ almost everywhere (**Figure 5B**, green line), reflecting the fact that the simplified 1D model (**Figure 5A**, green dashed line) semi-quantitatively predicts the applied force. Point $b$ in **Figure 5A** marks the onset of the high-strain regime. Here the flatness $e$ measurably deviates from its undeformed value. Noticeably, $e$ also varies along the filament contour length, attaining largest distortions at the filament midpoint (**Figure 5B**, blue line line). The significant cross-sectional deformations in the high-strain regime demonstrate that the non-classical behavior of force with compressive strain in microtubules isindeed due to cross-sectional flattening. The location of the force plateau is highly sensitive on the circumferential Young's modulus, $E_c$, and significantly less on the other parameters. Increasing $E_c$ shifts the onset of the plateau towards larger compressive strains, eventually leading to its disappearance.

It is well-established that local cross-sectional buckling (kinking) can occur in a thin-walled tube because of cross-sectional ovalization (**Huang et al., 2017**). This is indeed what we observe - the cross-section is circular (**Figure 5B**, green and blue curves) until a critical bending moment is reached. The blue curve in **Figure 5B** represents the profile right before the local buckling transition; afterwards, a sharp increase in $e$ takes place at the middle, representing the development of a kink. With further compression, subsequent kinks can develop - one at a time - in its vicinity. The red curve in **Figure 5B**, for example, shows five kinks (visible as spikes in the flatness profile) in the microtubule configuration corresponding to the point labeled es$c$ in **Figure 5A**. The inset in **Figure 5B** shows that the corresponding curvature profiles effectively mirror the flatness profiles in the main figure. Thus, discontinuities in the curvature profile of a filament signal the existence of cross-sectional ovalization and eventual kinking. In simulations, the formation of kinks can be identified directly from the force-strain plots as a small drop in the buckling force (**Figure 5A**, red). Indeed, hysteresis associated with forward and backward compressionis observed in simulations in which the filament forms kinks. The often observed persistence of hysteresis at very low strains, as seen in **Figures 2B**, **4** and **5**, is likely a simulation artefact, due to the lattice getting trapped in a local minimum; the addition of

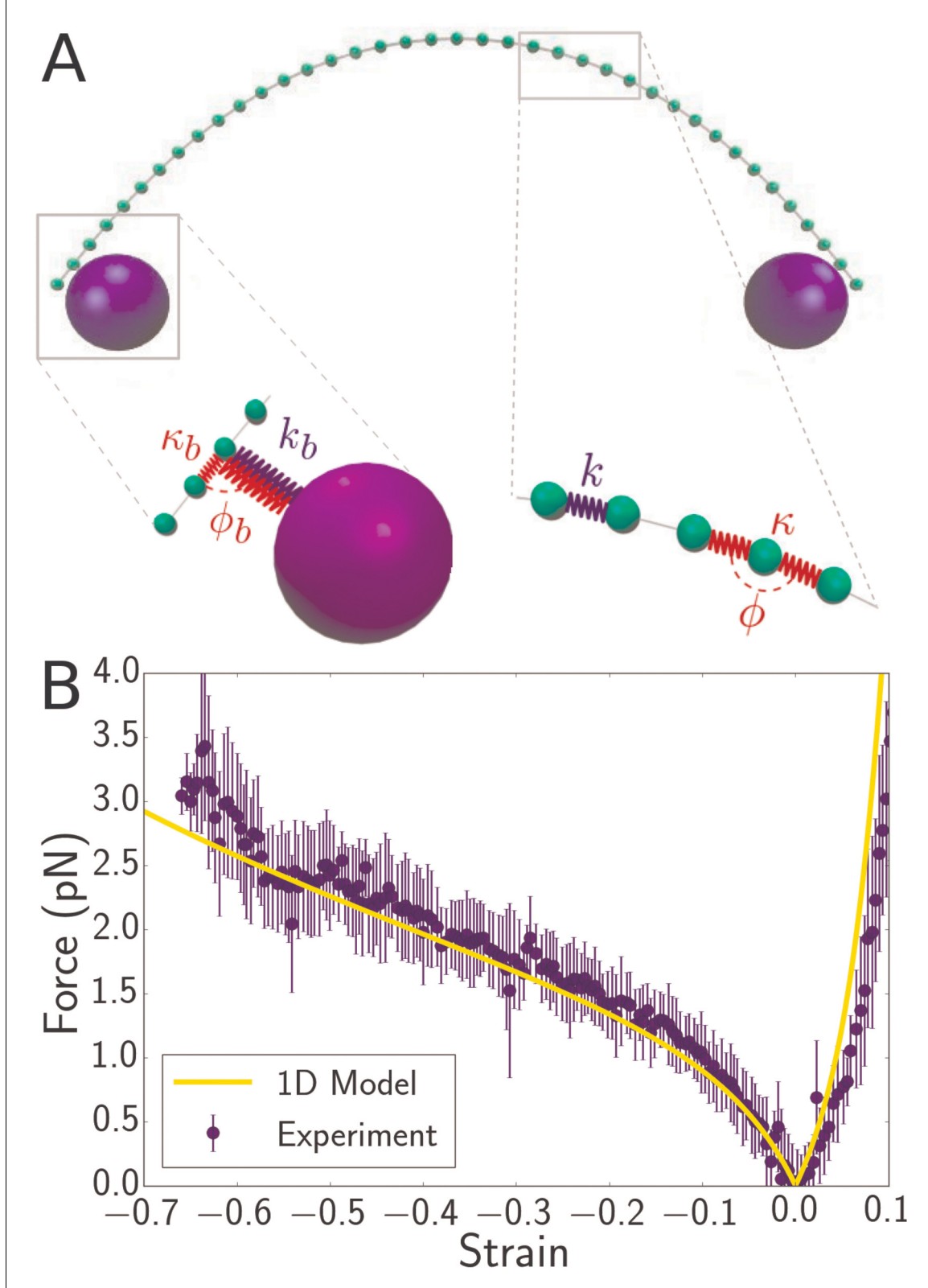

**Figure 3.** Discrete mechanical model of bacterial flagellum. (**A**) Schematic of spring network model for bacterial flagellum. Green particles discretize the flagellum while blue particles represent the optical beads. Right inset: closeup of a region of the filament, showing that neighboring masses are connected by linear springs of stiffness $k$ which give rise to a stretching energy $V_{\text{stretch}}(k, l, l_0)$ according to **Equation 2**, where $l_0$ is the stress-free length of the springs. In addition, each triplet of neighbors determines an angle $\phi$ which dictates the bending energy of the unit: $V_{\text{bend}}(\kappa, \phi, \pi)$ (**Equation 3**).
*Figure 3 continued on next page*

*Figure 3 continued*

Left inset: closeup of the region where the bead connects to the flagellum. The bead particle is connected to a single particle of the flagellum by a very stiff spring $k_b$, of stress-free length $b$ corresponding to radius of the optical bead. In addition, there is a bending interaction $V_{\text{bend}}(\kappa_b, \phi_b, \pi/2)$, where $\phi_b$ is the angle determined by the bead, its connecting particle on the flagellum, and the latter's left neighbor. (B) Force-strain curve from experiment (*blue*) and simulation (*yellow*) for a flagellum of length $L = 4.1\ \mu m$. Errorbars are obtained by binning data spatially as well as averaging over multiple runs. The bending rigidity obtained from the fit is $B = 4\,pN \times \mu m^2$. The classical buckling force for a rod of the same length and bending rigidity with pinned boundary conditions is $F_c \approx 2.3\text{pN}$.

DOI: https://doi.org/10.7554/eLife.34695.007

thermal noise either eliminates hysteresis or eliminates its low-strain persistence. Experimental data is too noisy to draw any conclusions regarding the presence or absence of hysteresis.

## Critical curvature is relatively small

The critical bending moment $M_B$ at which a hollow tube becomes unstable, leading to local collapse and kinking, is given by (*Brazier, 1927*):

$$M_B \approx \frac{2\sqrt{2}\pi}{9} h^2 R E, \tag{5}$$

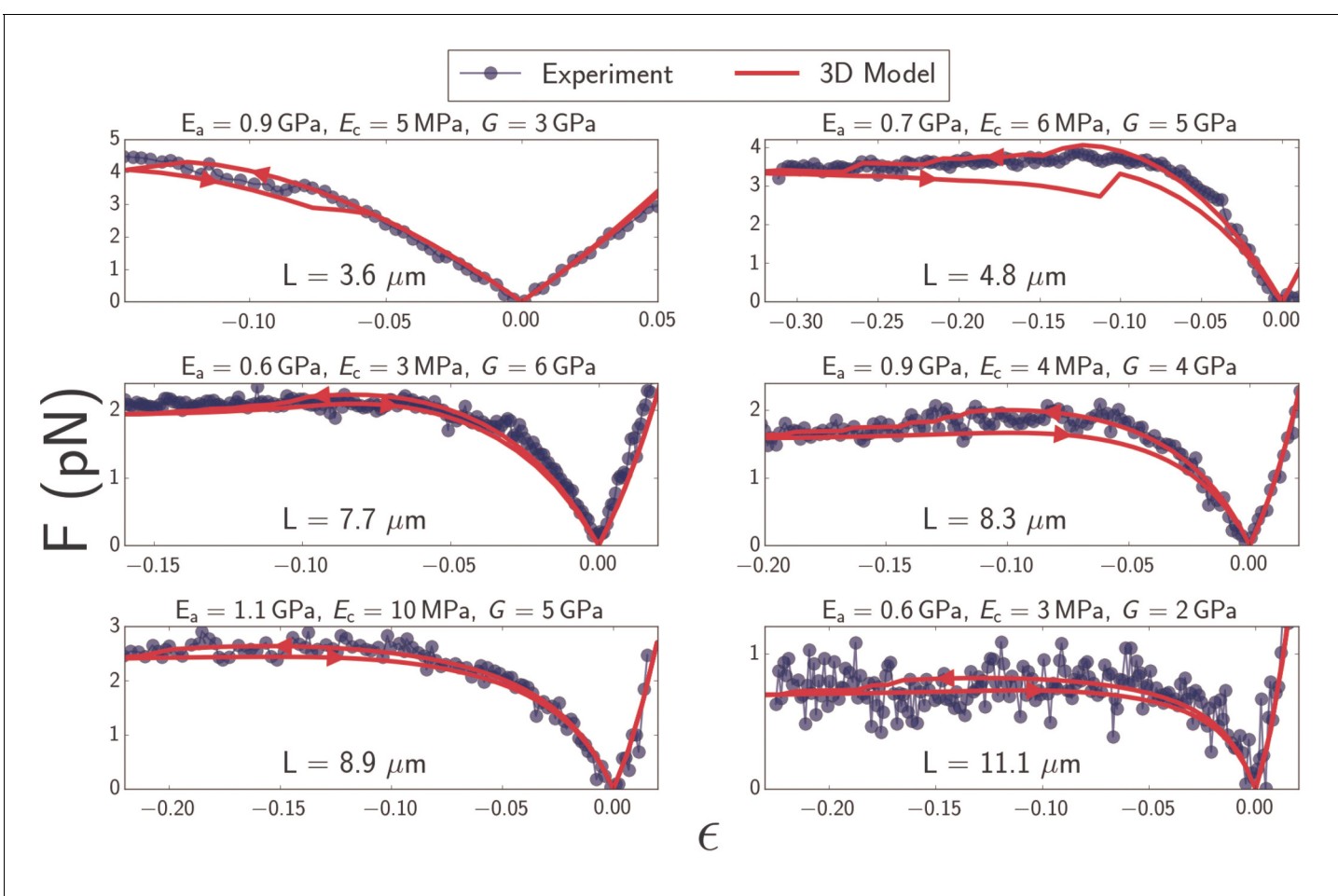

**Figure 4.** Plots of force $F$ as a function of strain $\epsilon$ (Equation *1*) for microtubules ranging in length from 3.6 to $11.1\mu m$ (*blue*). Best-fit curves from simulation of the 3D model (*red*) and fit parameters (top) are shown for each plot. Arrows on the red curves indicate forward or backward directions in the compressive regime.

DOI: https://doi.org/10.7554/eLife.34695.009

**Table 2.** Range of macroscopic elastic parameters that fit experimental data for microtubules and flagella.

| System | Fit parameters | Range that fits data |
|---|---|---|
| | $E_a$ | 0.6–1.1 GPa |
| Microtubule | $E_c$ | 3–10 MPa |
| | $G$ | 1.5–6 GPa |
| Flagellum | $B$ | 3–5 $pN \mu m^2$ |

DOI: https://doi.org/10.7554/eLife.34695.010

where $h$ is the tube thickness, $R$ is the radius, and $E$ is its elastic modulus. Brazier's result applies to isotropic, infinitely long tubes. It has been extended to orthotropic materials (*Huang et al., 2017*) by replacing $E$ with the geometric mean of $E_a$ and $E_c$ and to finite-length tubes by including a numerical correction factor (*Takano, 2013*), so that we expect the following scaling for $M_B$:

$$M_B \sim h^2 R \sqrt{\frac{E_a E_c}{1 - \nu_a \nu_c}} \sim h^2 R \sqrt{E_a E_c}. \tag{6}$$

To understand this result, we first carry out simulations after subtracting the energy due to shear deformations and compute a 'modified' force as the derivative of all energies except for the shear energy. This modified force is then used to calculate a modified, shear-agnostic, critical bending moment. Multiplying force computed this way by the maximum vertical displacement of the filament yields the value of $M_B$ observed in simulation (*Figure 6*, vertical axis), which we compare to values of $M_B$ computed according to *Equation 6* using a numerical factor of unity (*Figure 6*, horizontal axis). We confirmed the scaling predicted in *Equation 6* by running additional simulations, varying the length of the microtubule $L$, their elastic moduli $E_a$, $E_c$, and $G$, and the size of the trapped bead beyond the typical range of our experiments (see *Figure 6*).

Our experimental results (*Figure 6*, stars) suggest that the critical bending moment is generally in the range $M_B \approx$ 1000–2000 pN nm. This approximate range of the critical bending moment enables us to estimate the critical curvature at which softening occurs (as curvatures may be more easily extracted from experiments). Upon Brazier buckling, the curvature profile spikes at the location of the kink(s), just like the flatness profile (*Figure 5B*, inset), showing the correspondence between critical curvature and critical bending moment. Values of $M_B$ of around 1000–2000 pN nm imply a critical curvature for the onset of softening on the order of 0.1 rad / $\mu$m, given a bending rigidity $B$ of ~ 10–20 pN $\times \mu m^2$. This estimate is confirmed by simulations, which show that the critical curvature at the onset of the Brazier buckling is close to 0.2 rad / $\mu$m for seven of eight experimental datasets (and 0.3 for the other dataset). Such values are small enough that they are likely reached and exceeded at least in some systems both in vivo, as in the beating flagella of *Chlamydomonas* (*Sartori et al., 2016*; *Geyer et al., 2016*) and in vitro, as in microtubule rings observed in gliding assays (*Liu et al., 2011*).

## Discussion

Using optical trapping we have determined the mechanical properties of microtubules by measuring how the strain changes with an applied extensile/compressive force. We found that the force required to buckle the filament levels off or decreases with increasing compressive strain, demonstrating that microtubules significantly soften above a critical strain. Such non-classical behaviors are quantitatively captured by overdamped molecular dynamics simulations of an orthotropic cylindrical shell model. The simulations reveal that microtubule softening is a consequence of cross-sectional deformations enabled by a small circumferential Young's modulus of a few MPa.

### MT softening could explain large variation of bending rigidity results

While it is known that MTs do not behave as isotropic Euler-Bernoulli slender rods - since, for instance, lateral bonds between adjacent protofilaments are much weaker than longitudinal bonds along protofilaments (*Nogales et al., 1999*; *VanBuren et al., 2002*; *Huang et al., 2008*) -many

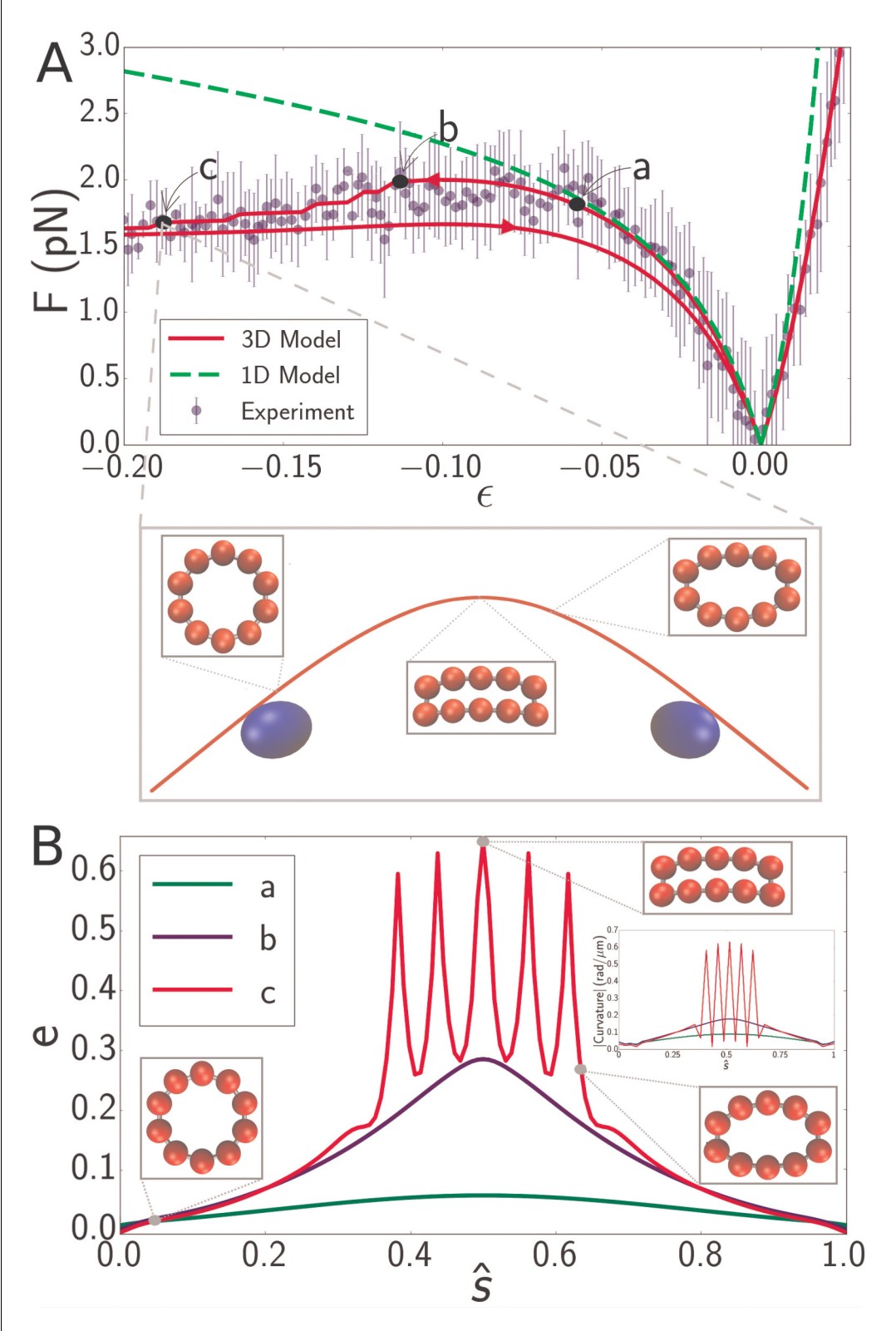

**Figure 5.** (A) Force-strain curve for a microtubule of length 8.3 $\mu m$ from experiment (*blue*) and simulation using a 1D model (*green, dashed*) or a 3D model (*red*). Arrows on the red curve indicate forward or backward directions in the compressive regime. Errorbars are obtained by binning data spatially as well as averaging over multiple runs. The fit parameters for the 3D model are $E_a = 0.9$ GPa, $E_c = 4$ MPa, and $G = 4$ GPa. The bending rigidity for the 1D model fit is $B = 23 pN\mu m^2$. Labels *a*, *b*, and *c* indicate specific points in the 3D simulation model. (inset) Snapshot from simulation

*Figure 5 continued on next page*

*Figure 5 continued*

corresponding to the point labeled c. Smaller insets show the shape of the cross-section at indicated points. (**B**) Profiles of flatness $e$ (*Equation 4*) along the dimensionless arclength coordinate $s = s/L$ (see *Figure 1A*) at points identified in panel A as a (green), b (blue), and c (red). Three insets show the shape of the cross-section at indicated points for the curve labeled c (*red*). A fourth inset shows the curvature profiles corresponding, by color, to the flatness profiles in the main figure.

DOI: https://doi.org/10.7554/eLife.34695.011

groups have used the Euler-Bernoulli beam model to interpret their data (*Kikumoto et al., 2006*; *Brangwynne et al., 2006*; *Li, 2008*; *Takasone et al., 2002*; *Jiang and Zhang, 2008*). Introducing a shear degree of freedom to the Euler-Bernoulli beam results in a Timoshenko beam model. This model remains inadequate, however, since it does not allow for anisotropy between the axial and circumferential directions, which is substantial and plays an important role in the mechanical behavior of MTs (*Deriu et al., 2010*; *Huang et al., 2008*; *Tuszyński et al., 2005*). Clearevidence supporting this assertion comes from experiments in which microtubules subject to osmotic pressure buckle radially at very low critical pressures (~600 Pa) – over four orders of magnitude lower than what is expected for an isotropic shellwith $E \sim 1$ GPa (*Needleman et al., 2005*). More sophisticated models of MTs as anisotropic elastic cylindrical shells allow for an additional degree of freedom in the radial direction and can be found in papers that examine MT persistence length (*Sim and Sept, 2013*; *Deriu et al., 2010*; *Ding and Xu, 2011*; *Gao et al., 2010*), oscillation modes (*Kasas et al., 2004*), response to radial indentation (*Huang et al., 2008*), or buckling under osmotic pressure (*Wang et al., 2006*).

Despite numerous studies on microtubule mechanics, considerable disagreement continues to persists regarding their coarse-grained elastic properties, including their bending rigidity or shear modulus (*Hawkins et al., 2010*). It is even unclear whether bending rigidity depends on the MT length or not (see, for example, [*Pampaloni et al., 2006*; *Kis et al., 2002*; *Deriu et al., 2010*; *Zhang and Meguid, 2014*] versus [*Gittes and Schmidt, 1998*; *Kikumoto et al., 2006*; *Van den Heuvel et al., 2008*]). Some of the disparity in measured elastic properties can be attributed to the variations in theexperimental protocols that could affect density of defects and filament heterogeneity as well as the number of microtubule protofilaments (*Zhang and Meguid, 2014*). Furthermore, different methods of stabilizing microtubules could also affect their coarse-grainedproperties. However, our work suggests that a significant source of variation could arise for a more fundamental reason, especially when interpreting results on mechanically deformed filaments. This is because theoretical models used so far to analyze properties of mechanical MT experiments may lack sufficient complexity to accurately describe all deformation regimes, and experiments that rely on the mechanical interrogation techniques have rarely if ever simultaneously explored the behavior of filaments in both large and small deformation regimes. In such a case, the force-strain curves are frequently measured only at high strains and thus leads to significant errors when extrapolated across all strain regimes using overly-simplistic Euler-Bernoulli or Timoshenko beam models. This methodology can significantly underestimate the microtubule persistence lengths that are measured using applied mechanical deformations. Indeed, a comprehensive summary of experimental results on MT flexural rigidity (*Hawkins et al., 2010*) strongly supports this interpretation: for example, the average flexural rigidity of Taxol stabilized MTs found via thermal fluctuations is more than twice as large as the average rigidity found from force-based experiments.

## Large shear modulus likely reflects physical deformation of tubulin units

Fitting experimentally measured force-strain curve to our theoretical model provides an estimate of the microtubule elastic moduli $E_a$, $E_c$, and $G$ (*Table 2*), which can be compared to values reported in the literature. The range of axial Young's moduli $E_a$ that fits our experiments is 0.6–1.1 GPa, consistent with values of 0.5–2 GPa reported previously in the literature (*Deriu et al., 2010*; *Enemark et al., 2008*; *Wells and Aksimentiev, 2010*; *Sept and MacKintosh, 2010*). Similarly, we find $E_c$ between 3–10 MPa, which is consistent with figures of a few MPa reported by some groups (*Wang et al., 2006*; *Tuszyński et al., 2005*), though other groups also report $E_c$ on the order of 1 GPa (*Zeiger and Layton, 2008*; *Schaap et al., 2006*). The discrepancy can be reconciled by noting that small radial deformations only impactthe weak bridges between adjacent protofilaments, while

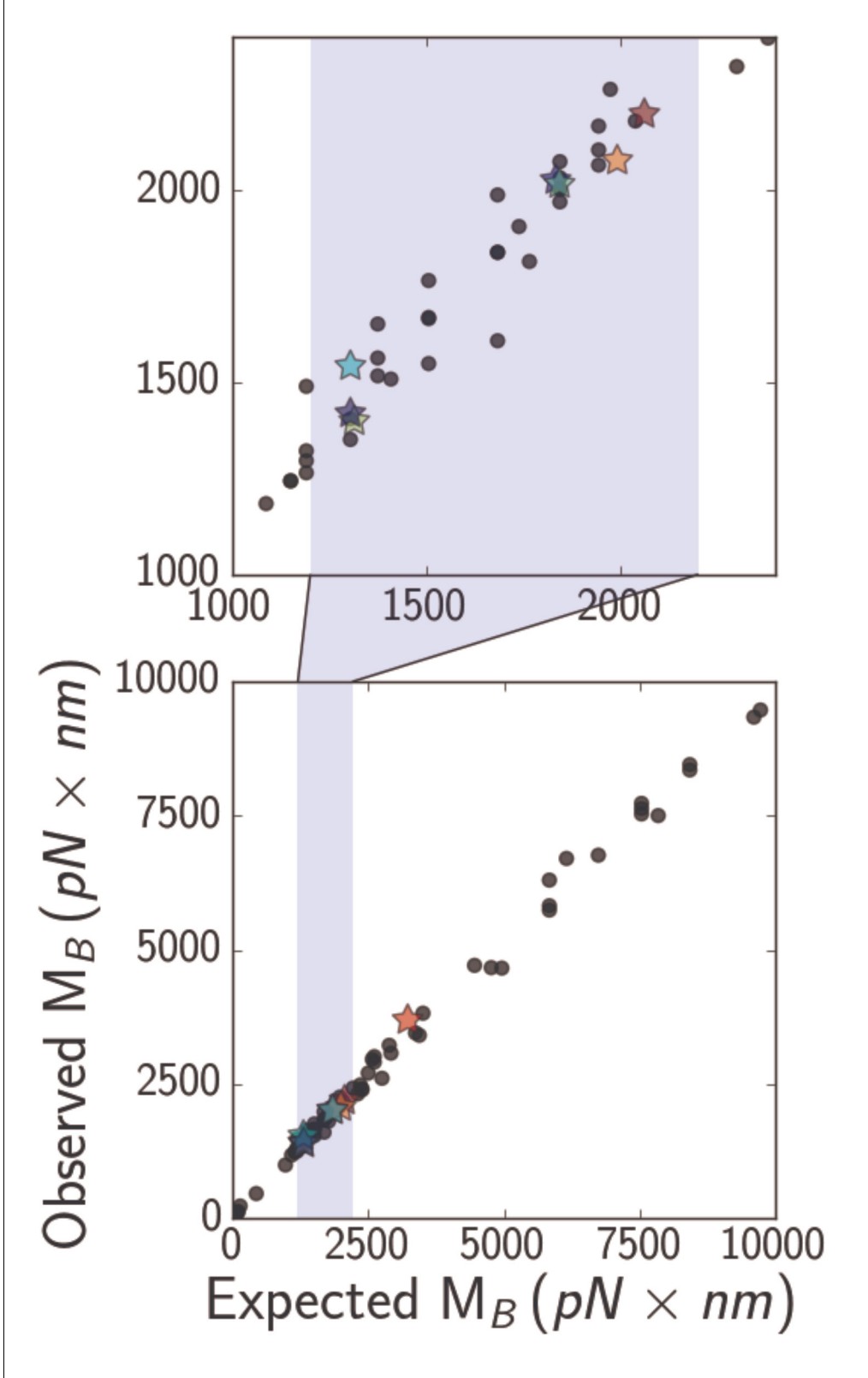

**Figure 6.** (*bottom*) Expected (*Equation 6*, with a numerical factor of unity) versus observed critical bending moments $M_B$ of simulated microtubules of various lengths, elastic moduli, and optical bead sizes. Data points that correspond to actual experimental parameters are marked with stars, while the rest are marked with black circles. The shaded region is zoomed-in on (*top*), to show that most experimental points fall in the range about 1000–2000 pN × *nm*.

*Figure 6 continued on next page*

*Figure 6 continued*

DOI: https://doi.org/10.7554/eLife.34695.012

larger ones may deform the tubulin dimers, which, like typical proteins, have a high stiffness in the GPa range (*Huang et al., 2008*; *Pampaloni et al., 2006*; *Needleman et al., 2005*).

We also extract the microtubule shear modulus $G$, which assumes values of a few GPa (*Table 2*), which is significantly larger than values reported in the literature (which can vary by as much as five orders of magnitude) from around 1 (*Pampaloni et al., 2006*), to 1 MPa (*Kis et al., 2002*), up to 100 MPa (*Deriu et al., 2010*; *Ding and Xu, 2011*; *Sept and MacKintosh, 2010*). This is surprising, since it is well-established that microtubule lateral bonds are very compliant; a small shear modulus is believed to potentially help the microtubule correct defects in assembly by shifting the offset between protofilaments during the nucleation phase (*Sim and Sept, 2013*). However, inter-protofilament bonds may be very compliant for small deformations (on the 0.2 nm scale), while larger deviations may approach the limit of physically deforming the tubulin units (*Pampaloni et al., 2006*). Following estimates in (*Pampaloni et al., 2006*), a strain on the order of $-0.01$ would lead to deformations on the 0.2 nm scale, while our experiments routinely go up to much larger strains around $-0.2$. Therefore, it's likely that the large values we obtained for shear modulus are due to large shear deformations that surpass the elastic limits of inter-protofilament bonds.

Notably, sufficiently large deformations can induce (reversible) breaking of lateral contacts between protofilaments and eventually even of longitudinal contacts between subunits in a protofilament as observed in radial indentation simulations (*Kononova et al., 2014*; *Jiang et al., 2017*) and corroborated with scanning force microscopy experiments (*Schaap et al., 2004*). Therefore, contact breaking may constitute an alternate or co-occurring reason for the large shear modulus obtained from fitting our experimental data. Thisis supported by comparable values of bending rigidity emerging from radial indentation simulations in (*Kononova et al., 2014*) (25 pN $\times \mu\mathrm{m}^2$) and from the current paper (10–20 pN $\times \mu\mathrm{m}^2$ where flattening and buckling become noticeable).

## Softening may occur in parameter ranges typical of typical biological systems

Our experimentally measured force-strain curves imply that the critical curvature for the onset of Brazier buckling is relatively small, around 0.2 rad $/\mu\mathrm{m}$. Such curvatures are frequently observed in both in vitro and in vivo systems. For example, long-lived arcs and rings observed in microtubule gliding assays (*Liu et al., 2011*; *Bourdieu et al., 1995*) have curvatures between 0.4–2 rad $/\mu\mathrm{m}$. From a different perspective, microtubules are the essential structural motif of diverse non-equilibrium materials including active isotropic gels, nematic liquid crystals, motile emulsions and deformable vesicles (*Sanchez et al., 2012*; *Keber et al., 2014*). The rich dynamics of all these systems is driven by the buckling instability of microtubule bundles that is driven by internal active stress generated by kinesin motors. In some systems the curvature associated with this buckling instability is large enough for microtubule softening to be relevant.

In living organisms, the static curvatures of the quiescent eukaryotic flagella, such as those found in *Chlamydomonas Reinhardtii*, are about 0.25 rad $/\mu\mathrm{m}$ (*Sartori et al., 2016*) while the oscillating dynamic component can increase the curvature up to about 0.6 rad $/\mu\mathrm{m}$ (*Geyer et al., 2016*), although presence of numerous microtubule associated proteins could significantly alter filament's mechanical properties. Furthermore, diverse active processes within a cellular cytoskeleton are also capable of generating highly curved microtubule configurations (*Brangwynne et al., 2006*). We may thus speculate about the biological significance of MT softening, with one hypothesis being that it decreases MT susceptibility to mechanical failure or depolymerization (*Mohrbach et al., 2012*).

The majority of other previous experiments have only examined the properties of stabilized microtubules which do not polymerize/depolymerize on relevant timescales. In comparison, a recent significant advance examined mechanical properties of dynamical microtubules that coexist with a background suspension of tubulin dimers (*Schaedel et al., 2015*). In particular, this study demonstrated that repeated large-scale deformations locally damage microtubules, which leads to effectively softer filaments. This damageis accompanied by a loss of tubulin monomers. Furthermore, after the external force ceases the damaged filaments effectively self-repair, as the tubulin

monomers from the background suspensions incorporate back into the damaged regions. In comparison, here we study stabilized filaments that do not show any aging phenomena. However, it seems plausible that the ovalization and the formation of kinks is also relevant to dynamical microtubules, and the regions of high strain might be the location where monomers preferentially dissociate from filament.

It is also worth noting that the softening of microtubules we observe is analogous to phenomena that have already been observed and quantified in carbon nanotubes (e.g. see reviews by [*Thostenson et al., 2001*; *Wang et al., 2007*]). In particular, experiments demonstrate the formation of a single kink, followed by a multiple kink pattern upon further bending (*Iijima et al., 1996*), while simulations predict a reduction in effective nanotube stiffness, as the buckling force drops and remains almost constant after kinking eYakobson1996. Hysteresis due to plastic deformations triggered by kinking events has also been observed, at least in the case of multi-walled carbon nanotubes (*Jensen et al., 2007*). Stiffness variations of carbon nanotubes arise from deformation modes which cannot be explained by simple Euler-Bernoulli or Timoshenko rod models.

In conclusion, we have found that micron-long microtubules subject to buckling forces of a few pN become more mechanically compliant at relatively low strains due to significant cross-sectional deformations and subsequent buckling. This result seems biologically relevant, as the critical curvatures for the softening transition are surpassed, for instance, in the beating of *Chlamydomonas* flagella. Additionally, the softening of MTs with increasing strain provides an explanation for the discrepancy between values of flexural rigidity inferred by passive (thermal fluctuations) versus active methods, since the latter typically access higher strains and thus infer effectively lower rigidities. Despite the complex nature of microtubules, we found that their mechanical properties - at least as concerning buckling experiments via optical trapping - can be quantitatively summarized by three elastic moduli, even for very large strains, and need to be accounted for to explain observations that arise naturally in many biological systems.

## Materials and methods

### Tubulin and microtubule polymerization

Tubulin was purified from bovine brain tissue according to the established protocol (*Castoldi and Popov, 2003*). We conjugated two fluorescent dyes: Alexa Flour 568 NHS Ester (Life Technologies, A-20006), Alexa Flour 647 NHS Ester (Life Technologies, A-20003) orbiotin-PEG-NHS (Thermo Scienti1c, 20217) to tubulin as described previously (*Hyman et al., 1991*). We prepared NEM-modified tubulin by incubation of unmodified dimers at a concentration of 13 mg/ml with 1 mM NEM (N-Ethylmaleimide, Sigma, E3876) and 0.5 mM GMPCPP (Jena Bioscience, NU-405S) on ice for 10 min and then quenching the reaction with 8 mM beta-mercapthoethanol (Sigma, M6250) for another 10 mins (*Hyman et al., 1991*). All labeled and unlabeled tubulin monomers were stored at $-80°$C.

### Bead functionalization

Carboxyl silica microspheres (d = 0.97 $\mu$m, Bangs Labs SC04N/9895) were coated with NeutrAvidin-DyLight488 conjugate (Thermo Scientific, 22832) by incubating the mixture of beads and protein at pH = 8.0 in presence of N-Hydroxysuccinimide (NHS, Sigma, 130672) and N-(3-Dimethylamino-propyl)-N'-ethylcarbodiimide hydrochloride (EDC, Sigma, E6383). The use of fluorescently labeled protein allows the bead to be imaged with fluorescence microscopy in order to verify Neutravidin presence on the surface. However, this is not always desirable. We have tested the protocol with a number of Neutravidin and Streptavidin proteins from various suppliers (Sigma, Thermo Fisher) and concluded that the protein attachment to the surface is reliable and reproducible.

### Buffer solutions

Microtubules (MTs) were polymerized in the M2B buffer that contained 80 mM PIPES (Sigma, P6757), 2 mM MgCl 2, 1 mM EGTA (Sigma, E3889) and was titrated to pH = 6.8 with KOH. As a result of titration with potassium hydroxide, the buffer contained 140 mM of K + ions. In order to extend the range of accessible ionic strength in the solution, we used M2B- 20 buffer that contained 20 mM PIPES, 2 mM MgCl 2, 1 mM EGTA and was titrated to pH = 6.8 with KOH. As a result of titration with potassium hydroxide, the buffer contained ~35 mM of K + ions. Oxygen scavenger

solution was prepared immediately before use by combining equal volumes of solutions of glucose (Sigma, G7528), glucose oxidase (Sigma, G2133) and catalase (Sigma, C40). Resulting mixture was dilutedinto the final sample in order to achieve the following concentrations: 40 mM Glucose, 250 nM glucose oxidase, 60 nM catalase (*Gell et al., 2010*).

## Microscopy and optical trapping

The position of the detection bead is sampled with the QPD at a rate of 100 KHz and averaged in bins of 1000 samples, effectively resulting in a sampling rate of 100 Hz, that is about 100 times per step. The force on the detection bead is obtained by multiplying the displacement of the bead by the stiffness $k_o$ of the optical trap.

All experiments required simultaneous use of fluorescence and brightfield microscopy modes, as well as optical trapping. We used an inverted Nikon Eclipse TE2000-U with Nikon PlanFlour 100x oil-immersion objective (NA = 1.3) equipped with the epifluorescence illumination arm, either a mercury-halide (X-Cite 120Q) or LED (Lumencor SOLA) light source, and an appropriate set of filter cubes (Semrock). In order to have multiple independently controlled optical traps we used an optical bench set-up based on two acousto-optical deflectors (AODs) - see (*Ward et al., 2015*). Briefly, the AOD consists of a transparent crystal, in which an acoustic wave is inducedby applied radio-frequency (RF, several MHz) electric current. The incoming laser beam interacts with the traveling sound wave and is deflected at a small angle, which depends on the RF current frequency. When the AOD crystal is imaged onto the back focal plane of a microscope objective, deflection of the beam is converted into translation of the focused laser spot in the focal (specimen) plane. By combining two AODs with perpendicular orientation, one can achieve precise control of the laser trap position in two dimensions within the focal plane of the microscope Molloy1997.

In our set-up, laser beam from a continuous infrared laser (1064 nm, Coherent Compass CW IR) is expanded and sent through the AOD (Intra-Action Corp., Model DTD-276HD2). The deflected beam is sent through a set of telescope lenses into the objective. The deflection angle of the AOD is controlled by custom LabVIEW software, which allows for positioning of optical traps in the x-y plane with nanometer precision.

In order to calibrate the optical traps, the setup was equipped with a separate laser (830 nm, Point Source iFlex 2000) and a quadrant photodiode (QPD) detection system (*Simmons et al., 1996*). Voltage readings from the QPD, which are directly proportional to the bead x and y coordinates, were obtained with 100 kHz frequency. The voltage readings were Fourier transformed and the trap stiffness was obtained from the Lorentzian fit to the power spectrum (*Berg-Sørensen and Flyvbjerg, 2004*; *Gittes and Schmidt, 1998*)

## Bead attachment

We follow bead-microtubule attachment protocols very similar to those in the literature (*van Mameren et al., 2009*; *Kikumoto et al., 2006*). We prepare segmented microtubules in which short regions (seeds) are labeled with biotin. The biotinylated seeds and the biotin-free elongated segment are labeled with two different fluorescent dyes, which allows us to distinguish them and attach the Neutravidin-coated silica beads to the biotin labeled MT seeds through manipulations with laser tweezers (*Figure 1A*). Presence of biotin either throughout the filaments or in localized segments does not affect the measured single filament buckling force. For the case of the bacterial flagella, we labeled the entire filament with the fluorescent dye and biotin. In the end, our experiments seem consistent with an essentially rigid microtubule-bead attachment. We note that a finite range of bead movement or rotation relative to the attachment point would appear as a flat region near zero strain in the force-strain curve, which we do not observe.

## Force averaging

Trap separation is controlled by acousto-optical deflectors (AODs), and changed in discreet steps of 2–10 nm every 0.5–1 s. For a given trap separation, the force on the detection bead is obtained by multiplying the displacement of the bead (from the center of the optical trap) with the stiffness of the calibrated optical trap. The position of the detection bead is sampled with a quadrant photodiode (QPD) at a rate of 100 KHz and averaged in bins of 1000 samples, effectively resulting in a sampling rate of 100 Hz. Since trap separation is changed every 0.5–1 s, the force on the bead is

measured 50–100 times at a given distance. Several cycles of buckling and extension ("back-and-forth runs') are performed for a given filament so that in the end, the raw force data is binned by trap separation and the average and standard deviation of the force in each bin are computed. Notably, then, force is averaged both over time and over multiple runs.

The minimum in the force-displacement curve separates the tensile and the compressive regimes and allows us to determine the equilibrium bead separation of the beads, that is the effective length $L$ of the filament segment being actively stretched or buckled. Due to thermal fluctuations and other noise, experimentally measured forces fluctuate in direction (and magnitude). Since fluctuations in the direction perpendicular to the line connecting the optical traps average out to zero, the forces reported in our manuscript are the parallel components. In the manuscript, we present force - averaged and projected as explained - in terms of strain corresponding to bead separation, rather than optical trap separation (the schematic in **Figure 1A** illustrates the distinction). We compute bead separation from trap separation, measured force, and optical trap stiffness.

## Simulations of flagellar buckling

To quantitatively describe the measured force-strain curve, the flagellar filament is modeled as a collection of $N$ masses connected to their neighbors with very stiff linear springs of constant $k$ (**Figure 3A**, right inset, blue). Bending energy of a triplet is encoded by a parameter $\kappa$, and $\phi$, the angle determined by the triplet (**Figure 3A**, right inset, red) according to **Equation 3**. The trapped beads are represented as masses connected to their attachment points by springs of stiffness $k_b$ and rest length equal to the radius of the bead $b$. The optical beads are enforced to remain normal to the filament at the attachment point by defining a bending energy $V_{\text{bend}}(\kappa_b, \pi/2)$ with a sufficiently large $\kappa_b$ and rest angle $\pi/2$ for the triplet consisting of the bead, its attachment point, and a neighboring mass (**Figure 3A**, left inset).

Since the flagellum and the bead are essentially inextensible, $k$ and $k_b$ are set to values which are sufficiently large to essentially establish them as fixed parameters. The same is true for $\kappa_b$, as explained above, and for the number $N$ of beads used that make up the flagellum. Thus, the only free parameter is $\kappa$, which is related to flexural rigidity $B$ as follows:

$$\kappa = \frac{BN}{L}.$$

## Simulations of microtubule buckling

To fit experimental data for microtubules we model them as 3D networks of springs and simulate their quasistatic, overdamped mechanical response via the molecular dynamics software Espresso (**Arnold et al., 2013**; **Limbach et al., 2006**)

In vivo, MTs most commonly appear with 13 protofilaments (although there are exceptions depending on the cell type), whereas in vitro structures with 9–16 protofilaments have been observed (**Mohrbach et al., 2012**). Since the exact structure of microtubules can vary and since it has been shown that protofilament orientation is not important with respect to mechanical properties (**Hunyadi et al., 2007**; **Donhauser et al., 2010**), we consider the simplest case of 10 aligned protofilaments. Because microtubule monomers are composed of alpha- and beta-tubulin, they can contact other monomers laterally through alpha-alpha, beta-beta, or alpha-beta interactions. While there are differences in these interactions (and generally, in the mechanical properties of alpha- and beta-tubulin), they are relatively small so that it is safe to neglect these differences (**Sim and Sept, 2013**; **Zhang and Meguid, 2014**).

The microtubule is modeled as a planar spring network network wrapped into a cylindrical shell (**Figure 2A**). Energies are of two types: stretching energies $V_{\text{stretch}}$ (**Equation 2**) determined by the distance between pairs of particles connected by springs, and bending energies $V_{\text{bend}}$ (**Equation 3**) determined by the angle formed by neighboring triplets. Anisotropy is achieved by setting independent parameters $k_a$, $\kappa_a$ and $k_c$, $\kappa_c$ for the bending and stretching energies in the axial and circumferential directions, respectively (**Figure 2A**). The parameter $d$ (the distance between consecutive simulated dimers along the same protofilament) can set the level of coarse-graining, with $d \approx 8\,\text{nm}$ for real microtubules. The distance between dimers at the same cross-section is denoted by $p = 2R\sin(\pi/10)$.

*Table 1* shows the relationship between the microscopic parameters of the model and the three corresponding macroscopic elastic constants $E_a$, $E_c$ and $G$. The formulas are based on the orthothropic elastic shell model and involve an 'equivalent' thickness $h$ and a 'effective' thickness for bending $h_0$. The reason for this duality is that the actual thickness of the cross section of a microtubule varies periodically along the circumferential direction, between a minimum of about 1.1 nm (the 'bridge' thickness) and a maximum of about 4–5 nm (*Huang et al., 2008*). When modeled as a shell of uniform thickness, the actual cross section of a microtubule is replaced by an annular cross section with equivalent thickness $h \approx 2.7$ nm (*de Pablo et al., 2003*). However, the effective bending stiffness is different; since most of the strain is localized to the bridges between the protofilaments (*Schaap et al., 2006*), the effective bending stiffness is closer to the bridge thickness. According to (*de Pablo et al., 2003*; *Schaap et al., 2006*), $h_0 \approx 1.6$ nm.

It must be noted that microtubules are not exactly thin shells, while the formulas shown in *Table 1* apply in that limit. As a consequence, the numerical factors in the formulas are not precisely correct. Because of this, the formulas need to be adjusted by numerical factors; we do so by considering, for instance, a basic shear deformation, computing its elastic strain energy and adjusting the numerical factor to ensure agreement between the energy computed in the simulation and the theoretical energy. Interestingly, the orthotropic shell model does not contain an energy penalty for cross-sectional shearing. In simulations, this leads to an unphysical instability which can be resolved, for instance, by running the simulation at very high damping (which would, however, greatly increase the runtime). A faster way of resolving this artifact is to stabilize the microtubule in the lateral direction, effectively not allowing its centerline to break lateral symmetry by going out of plane.

Each optical bead is represented by a particle connected to a single particle on the microtubule by a very stiff spring as well as a very stiff angle interaction aimed at keeping the 'bead' perpendicular to the attachment point. The left bead is fixed, while the right one is free to move. Starting with the filament unstretched, we move the mobile bead in small increments $dx$. After each move, we fix the position of the bead and allow the system to relax. The relaxation time is (*Gittes et al., 1993*):

$$\tau_{\text{relax}} \sim \frac{\gamma L^4}{B},$$

where $\gamma$ is the friction coefficient. The parameter $\gamma$ determines both how fast the system relaxes (smaller $\gamma$ leads to faster relaxation) and how fast its kinetic energy decays (larger $\gamma$ leads to faster decay). Sufficiently large values of $\gamma$ correspond to overdamped kinetics, but the trade-off between relaxation and decay of kinetic energy implies that intermediate values of $\gamma$ optimize computation time. Such values of $\gamma$ are still acceptable, as they will lead to the same results provided kinetic energy has dissipated sufficiently and that there are no instabilities. To check that the value of $\gamma$ and the relaxation timescales are adequate, the system can be run forwards then backwards - if there is no hysteresis, then the parameters are adequate. This only holds until the system undergoes a kinking transition; once kinking takes place, hysteresis will occur even in the truly overdamped limit.

## Acknowledgements

The experimental portion of this work was primarily supported by the U.S. Department of Energy, Office of Basic Energy Sciences, through award DE-SC0010432TDD (FH and ZD) while the theoretical portion was supported by Harvard MRSEC through grant NSF DMR 14-20570. Experiments on bacterial flagella were supported by National Science Foundation through grant NSF-MCB-1329623 and NSF- DMR-1420382. We also acknowledge use of the Brandeis Materials Research Science and Engineering Center (MRSEC) optical and biosynthesis facilities supported by NSF-MRSEC-1420382

## Additional information

### Funding

| Funder | Grant reference number | Author |
| --- | --- | --- |
| U.S. Department of Energy | DE-SC0010432TDD | Feodor Hilitsk Zvonimir Dogic |

| National Science Foundation | NSF-MCB-1329623 | Zvonimir Dogic |
| National Science Foundation | DMR-1420382 | Zvonimir Dogic |
| Harvard MRSEC | NSF DMR 14-20570 | Edvin Memet L Mahadevan |

The funders had no role in study design, data collection and interpretation, or the decision to submit the work for publication.

## Author contributions
Edvin Memet, Conceptualization, Software, Formal analysis, Investigation, Visualization, Methodology, Writing—original draft, Writing—review and editing; Feodor Hilitski, Data curation, Software, Validation, Investigation, Methodology, Writing—review and editing; Margaret A Morris, Formal analysis, Investigation; Walter J Schwenger, Data curation, Software, Formal analysis; Zvonimir Dogic, Formal analysis, Funding acquisition, Project administration, Writing—review and editing; L Mahadevan, Conceptualization, Formal analysis, Supervision, Funding acquisition, Methodology, Project administration, Writing—review and editing

## Author ORCIDs
Edvin Memet (iD) http://orcid.org/0000-0001-9414-597X
Feodor Hilitski (iD) https://orcid.org/0000-0001-5629-1407
Zvonimir Dogic (iD) http://orcid.org/0000-0003-0142-1838
L Mahadevan (iD) http://orcid.org/0000-0002-5114-0519

## Decision letter and Author response
Decision letter https://doi.org/10.7554/eLife.34695.022
Author response https://doi.org/10.7554/eLife.34695.023

# Additional files

## Supplementary files
• Source Code 1. Source code files and source data for *Figure 6* and source code files along with instructions for generating the data in *Figure 2*.
DOI: https://doi.org/10.7554/eLife.34695.013

• Supplementary file 1. List of parameters used in the paper.
DOI: https://doi.org/10.7554/eLife.34695.014

• Supplementary file 2. (left) Summary of previous experiments examining microtubule stiffness, adapted from (*Hawkins et al., 2010*). (*right*) Box plot comparing bending stiffness values obtained via thermal fluctuations and mechanicalbending for microtubules stabilized with GDP Tubulin + Taxol (*orange*), GMPCPP Tubulin (*green*), and GDP Tubulin (*blue*)
DOI: https://doi.org/10.7554/eLife.34695.015

• Transparent reporting form
DOI: https://doi.org/10.7554/eLife.34695.016

## Data availability
Source data has been provided for Figure 6 along with source code files. Source code files have been provided for Figure 2 along with instructions for generating the data.

The following previously published dataset was used:

| Author(s) | Year | Dataset title | Dataset URL | Database, license, and accessibility information |
| --- | --- | --- | --- | --- |
| Hawkins T, Mirigian M, Yasar MS, Ross JL | 2010 | Flexural Rigidity of Single Microtubules. | https://doi.org/10.1016/j.jbiomech.2009.09.005 | The data used is available in the figures and tables of the manuscript |

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
