## [Decision Letter]

Thank you for submitting your article "Microtubules soften due to cross-sectional flattening" for consideration by *eLife*. Your article has been reviewed by three peer reviewers, and the evaluation has been overseen by a guest Reviewing Editor and Anna Akhmanova as the Senior Editor. The following individual involved in review of your submission has agreed to reveal his identity: Gregory M Alushin (Reviewer #3).

The reviewers have discussed the reviews with one another and the Reviewing Editor has drafted this decision to help you prepare a revised submission. As you will see only one set of additional experiment is suggested. Other comments can be addressed by clarification and discussion in the main text.

Summary:

The manuscript addresses a long-standing problem in microtubule mechanics, namely the discrepancy between the estimation of mechanical parameters (especially the bending rigidity) from different methods (passive - thermal fluctuations versus active - AFM or optical trapping).

Here the mechanical response of microtubules to bending forces has been investigated using an optical trap setup combined with coarse grained molecular dynamics simulations.

The main results are:

- Microtubule lattice experiences softening under relatively large compressive strains.

- This behaviour can be fitted with a new 3D microtubule lattice model which takes into account lattice anisotropy and its hollow cylindrical geometry. In contrast, the 1D elastic beam model is unable to reproduce the experimental behavior for microtubules except for the regime of very small (compressive or tensile) strains.

- An engineered bacterial flagellum more akin to a classical 1D beam, displays the expected Euler-Bernoulli bending mechanics, confirming that the microtubule behavior is not an experimental artifact. In this case the 1D model fits the experimental data very well.

Overall, the manuscript is clearly written, and the results are interesting. Presented data offer a new interpretation for the large variety in microtubule mechanical data measured by different methods and point at the key role of microtubule flattening in the modulation of their apparent stiffness, which is likely to significantly impact our understanding of microtubule mechanics and microtubule network architecture.

Essential revisions:

- Although the present data is potentially too noisy, a step experimental protocol could likely be employed to test the "aging" hypothesis. Specifically, a microtubule could be subjected to a sufficient force to buckle and soften it, followed by a low force. If the microtubule remains soft for some time after buckling, the time between the two force applications could be varied to establish the time constant for "healing". If this is feasible, including these experiments would substantially strengthen the study.

- The authors should explain how they control the attachment between bead and microtubule to obtain a configuration as shown in Figure 1A. The cartoon implies that the beads attach to the side of the microtubule in the direction of the curvature. Can the authors indicate the bead position in the experimental example in Figure 1B to make the cartoon in Figure 1A more credible? Indicated trap positions seem not consistent with the cartoon. Please describe in more detail, how the bead-microtubule attachment is controlled.

- Please describe in more detail how the force strain curve is obtained. For example, is there some averaging for the bead-trap distance, or bead-bead distance?

- From the simulations in Figure 2, Figure 4 and Figure 5 it seems that even at very low compressive strain < 0.05 a hysteresis is still present, and the microtubule may exist in two different conformations, associated with two different forces acting on the optical bead. Is this an artefact of the simulation algorithm, that the lattice gets trapped in a local minimum or do the authors have a physical explanation for the persistence of the hysteresis at very low strains? How is the steady state determined?

- In Figure 4, did the authors test experimentally compressive strains beyond 0.3? Did microtubules break and if so, at which point?

- It is unclear why the authors used a 10 protofilament model rather than a 13 protofilament one. Would the computational expense be so much greater in the latter case? One issue is that this choice leads to unrealistic lateral interactions/distances etc. And usually in vitro lattices with 10 protofilaments differ substantially from the 13 protofilament ones as they can exhibit more than the 1 seam from the 13 case. An explanation is needed.

- A clear comparison of the calculated with the experimentally measured microtubule shapes to validate the presence of the kinking instability is missing.

It seems that the model parameters are determined from a fit against the experimental force-strain curves. Could the authors show, e.g. in additional plots in Figure 5, how the experimental microtubule shapes compare to the simulated microtubule shapes at the points a,b, and c from Figure 5A?

- Alternative explanations could be envisaged and discussed:

Can the authors exclude a twisting of the microtubule lattice? The model calculations (Figure 5) show, that the microtubule lattice undergoes very drastic deformations in the kinking region. Is it plausible, that the microtubule lattice can accommodate these high strains without undergoing structural damage?

Can the authors exclude, that a lattice perturbation at the point of attachment between optical bead and microtubule is responsible for the observed softening?

- The mechanical lattice model is as simple as possible, which is a good thing since it limits the number of microscopic parameters. However, it is unclear whether a single shearing interaction (i.e. $\kappa_s$) in an effective 2D lattice is sufficient to stabilize the deformation of the lattice at the kinking instability. Furthermore, there seems to be some ambiguity concerning the relations between microscopic and macroscopic elastic constants (i.e. two different effective tube wall thicknesses are postulated, see. Table 2), which limits the quality of the predicted macroscopic elastic constants. This problem could be avoided by using for example a double shell lattice (with some diagonal elastic elements) which naturally includes stability against shear and bending in all directions.

---

## [Author Response]

- Although the present data is potentially too noisy, a step experimental protocol could likely be employed to test the "aging" hypothesis. Specifically, a microtubule could be subjected to a sufficient force to buckle and soften it, followed by a low force. If the microtubule remains soft for some time after buckling, the time between the two force applications could be varied to establish the time constant for "healing". If this is feasible, including these experiments would substantially strengthen the study.

Our model makes a number of specific predictions about the mechanical properties of microtubules. Most importantly it demonstrates that microtubules soften above a certain critical strain. Another prediction is that, in the absence of thermal noise, there is a hysteresis in the “forward” and “backwards” force-strain curve that may depend on the rate of strain change. (We find, however, that the addition of thermal noise generally eliminates hysteresis or at least eliminates its low-strain persistence.) Our paper unambiguously demonstrates the first effect. The second prediction is significantly more challenging to test experimentally, since most frequently the predicted jumps are of the same order of magnitude as the noise sources in our detection system. We have performed the force-extension experiments at multiple velocities but have failed to observe any statistically meaningful signature of hysteresis. The main limitation of this measurement lies in the stiffness of the optical trap and the various noise sources that introduce drift into our detection system. Serious efforts to identify the hysteresis steps would require a significant investment into the capabilities of our trapping setup, in order to track down and reduce all sources of detection error. The referees suggest an alternative method of looking for hysteresis, but it is not obvious for us that it would enhance our ability to detect it. We very much hope to pursue this question in the future studies once we upgrade our optical trap system.

- The authors should explain how they control the attachment between bead and microtubule to obtain a configuration as shown in Figure 1A. The cartoon implies that the beads attach to the side of the microtubule in the direction of the curvature. Can the authors indicate the bead position in the experimental example in Figure 1B to make the cartoon in Figure 1A more credible? Indicated trap positions seem not consistent with the cartoon. Please describe in more detail, how the bead-microtubule attachment is controlled.

All of the beads are attached to the side of a microtubule via a biotin streptavidin bond. We follow bead-microtubule attachment protocols very similar to those in the literature (van Mameren et al., 2009; Kikumoto et al., 2006). The former paper, in particular, contains a thorough description of the method. We bring an optically trapped bead to the microtubule side but the precise location of the bead attachment to the microtubule side is not exactly known.

In the end, our experiments seem consistent with an essentially rigid microtubulebead attachment. We note that a finite range of bead movement or rotation relative to the attachment point would appear as a flat region near zero strain in the force-strain curve, which we do not observe.

We know precisely the relative distance between the two trap positions. However, the relative position of the microtubule relative to traps is not precisely known. Therefore, circles in Figure 1B are drawn for illustration purposes and may not precisely represent actual trap positions, since neither beads nor traps are visible in fluorescence imaging. Moreover, bead positions would be practically indistinguishable from trap positions, since the maximum displacement of beads from the centers of their optical traps is on the order of 100 nm, corresponding to about one pixel in the image.

The uncertainty in the precise attachment of the bead to filaments is one of the main motivations for repeating our experiments with structurally simpler bacterial flagellar filaments. In both cases, our simulations include the model of the bead-filaments attachment described in Figure 1. With such a model we are able to quantitatively describe the buckling of the flagellar filaments with a simple one-dimensional elasticity. This provides strong evidence in support of rigid-side attachment of beads to filaments.

- Please describe in more detail how the force strain curve is obtained. For example, is there some averaging for the bead-trap distance, or bead-bead distance?

Trap separation is the independent variable, controlled by acousto-optical deflectors (AODs), and changed in discreet steps of 2 – 10 nm every 0.5 – 1 seconds. For a given trap separation, the force on the detection bead is obtained by multiplying the displacement of the bead (from the center of the optical trap) with the stiffness of the calibrated optical trap. The position of the detection bead is sampled with a quadrant photodiode (QPD) at a rate of 100 KHz and averaged in bins of 1000 samples, effectively resulting in a sampling rate of 100 Hz. Since trap separation is changed every 0.5 – 1 seconds, the force on the bead is measured 50 – 100 times at a given distance. Several cycles of buckling and extension (”back-and-forth runs”) are performed for a given filament so that in the end, the raw force data is binned by trap separation and the average and standard deviation of the force in each bin are computed. Notably, then, force is averaged both over time and over multiple runs.

Due to thermal fluctuations and other noise, experimentally measured forces fluctuate in direction (and magnitude). Since fluctuations in the direction perpendicular to the line connecting the optical traps average out to zero, the forces reported in our manuscript are the parallel components. In the manuscript, we present force – averaged and projected as explained – in terms of strain corresponding to bead separation, rather than optical trap separation (the schematic in Figure 1A illustrates the distinction). We compute bead separation from trap separation, measured force, and optical trap stiffness.

- From the simulations in Figure 2, Figure 4 and Figure 5 it seems that even at very low compressive strain < 0.05 a hysteresis is still present, and the microtubule may exist in two different conformations, associated with two different forces acting on the optical bead. Is this an artefact of the simulation algorithm, that the lattice gets trapped in a local minimum or do the authors have a physical explanation for the persistence of the hysteresis at very low strains? How is the steady state determined?

Video 2 illustrates the reason for the persistence of hysteresis at very low strains, showing that the simulated microtubule mostly but not fully recovers its circular cross-section on the backward run. However, as shown in the first panel of the figure below, we find that the addition of thermal noise greatly reduces or eliminates hysteresis, suggesting that, indeed, persistence of hysteresis in the noise-free simulations is due to the lattice getting trapped in a local minimum.

The simulation curves shown in the manuscript all correspond to a single back-and-forth run starting from the equilibrium position going into either compressive or extensive direction. It is a valid observation that due to the lattice being trapped in a local minimum, the buckling force in consecutive runs will generally fail to perfectly match the force measured in the previous runs until a steady state is reached. However, the discrepancy is bounded by hysteresis size, which is typically small, i.e. within experimental error bars.

- In Figure 4, did the authors test experimentally compressive strains beyond 0.3? Did microtubules break and if so, at which point?

In the experiment corresponding to Figure 2, we tested compressive strains up to almost 0.5 without any apparent breaking or irreversible damage of the microtubule. In other experiments, we did not go beyond a compressive strain of 0.3, since our focus was on doing multiple runs in the force plateau/decline region, which starts at significantly smaller strains (around 0.1). We have repeated our measurements hundreds of times and we observed noticeable microtubule breaking in just one instance, where microtubule exhibited a visible kick and the damage was irreparable. More typically, the first point of failure is the connection with the optical bead.

- It is unclear why the authors used a 10 protofilament model rather than a 13 protofilament one. Would the computational expense be so much greater in the latter case? One issue is that this choice leads to unrealistic lateral interactions/distances etc. And usually in vitro lattices with 10 protofilaments differ substantially from the 13 protofilament ones as they can exhibit more than the 1 seam from the 13 case. An explanation is needed.

Since it has been argued that protofilament orientation is not important with respect to mechanical properties (Hunyadi et al., 2007; Donhauser et al., 2010), our simulated microtubule does not exhibit, for instance, a seam; instead it is composed of straight/parallel protofilaments. In this simplified configuration, the precise number of protofilaments should not have a significant effect. The interaction strengths are scaled so as to correspond to the same macroscopic elastic moduli, regardless of the number of protofilaments (e.g. see Table 1, manuscript, for the relationship between *E_c_, κ_c_*, and *p*). We chose to work with 10 protofilaments initially to allow for slightly simpler troubleshooting/debugging of our simulations.

The reason is that the precise number of protofilaments in our simulation has little effect on the result (see Author response image 1). The longer, technical answer starts by noting, as pointed out in a separate question, that our single shell model is insufficient to stabilize the lattice at the kinkink instability (likely because the model does not contain an energy penalty for cross-sectional shearing). As a result, the cross-section essentially collapses completely at the location of the kink (in the absence of steric interactions), with the buckling force practically dropping to zero. This instability can be avoided by using very high damping (which becomes computationally expensive) or, equivalently, by fixing the positions of the’ middle’ (top and bottom) particles in each cross-section. This shortcut cannot be used with an odd number of protofilaments, as the symmetry plane of the cross-section passes through only one particle instead of two, which still allows for the instability to occur. Due to geometrical peculiarities, simulations with 12 PF (or 8, 16, and so on) are also difficult to stabilize without extremely high damping or additional parameters introduced through a second shell or diagonal springs. Simulations with 14 PF, however, agree well with results for 10 protofilaments (or 6, 18, and so on), though the rate of force decline in the’ plateau’ region is slightly higher for 14 PF. The discrepancy is due to the fact that the 14 PF cross-sections flatten somewhat more at kink locations than the 10 PF ones, reaching’ eccentricities’ (as defined in the manuscript) typically around 0.8 compared to 0.6 (see Author response image 1).

**Author response image 1. respfig1:** (A) Force-strain curve for a microtubule of length 7.7 μm from simulation using 10 protofilaments (red), 10 protofilaments with thermal noise (yellow), and 14 protofilaments (green). The fit parameters are E a = 0.6 GPa, E c = 4 MPa, and G = 6 GPa. B) Profiles of flatness e (Eq. 4, manuscript) along the dimensionless arclength coordinate ŝ = s/L (see Fig. 1A, manuscript) at pointsidentified in panel A with dots.

In conclusion, results are qualitatively robust between, say, 10 and 14 PF. Because of geometrical peculiarities arising from discretization, some choices for the number of protofilaments are difficult to stabilize (without extremely high damping) against cross-sectional shear instabilities.

- A clear comparison of the calculated with the experimentally measured microtubule shapes to validate the presence of the kinking instability is missing.It seems that the model parameters are determined from a fit against the experimental force-strain curves. Could the authors show, e.g. in additional plots in Figure 5, how the experimental microtubule shapes compare to the simulated microtubule shapes at the points a,b, and c from Figure 5A?

Simultaneous measurement of the force-strain curve and measurement of microtubule shape are generally not possible. The main reason is that microtubule filaments easily rotate along the axis connecting the two beads and are thus are most of the time out of the focal plane. While it is possible to keep the microtubule in the focal plane by applying a very slight flow, this comes at the expense of the ability to measure the buckling force. Other issues include the optical beads and the seed regions of the microtubule (labeled with a different dye) not being visible in fluorescence and the low resolution of experimental images.

For a few images where the filaments are seen to be in focus we have tried to fit the microtubule shape to theoretical predictions. For example, the filament in Figure 5 from the manuscript seems to be in focus at a strain around -0.21, near the point labeled (c) in that figure. Trying to match the contour from fluorescence imaging to the profile of the simulated filament at that strain reveals decent agreement but does not lead to perfectly compatible bead positions (Author response image 2). Similarly, trying to match the optical bead positions between simulation and experiment leads to non-matching filament shapes (Author response image 2). This can easily be attributed to the filament not being perfectly in the focal plane, but rather tilted at some angle. With this assumption, the shapes seem fairly consistent. However, due to the concerns raised above it seems to us that such analysis should not be included in the manuscript.

**Author response image 2. respfig2:** Comparison of shape between experiment (filament: (green); approx-imate trap positions: (blue)) and simulation (filament: (red); optical beads:(gray)) for a microtubule of length 8.3 μm (same as in figure 5, manuscript) at astrain ε ≈ −0.21 by A) aligning filament profiles and B) aligning bead positions(estimated).

- Alternative explanations could be envisaged and discussed:Can the authors exclude a twisting of the microtubule lattice? The model calculations (Figure 5) show, that the microtubule lattice undergoes very drastic deformations in the kinking region. Is it plausible, that the microtubule lattice can accommodate these high strains without undergoing structural damage? Can the authors exclude, that a lattice perturbation at the point of attachment between optical bead and microtubule is responsible for the observed softening?

The simulation does indeed exhibit a twisting mode in a parameter regime in which the shear modulus is small (order MPa) and the circumferential Young’s modulus is large (order GPa). The large value of the latter modulus, however, seemingly conflicts with the observation by Needleman et al. of cross-sectional collapse of microtubules at very low osmotic pressures. Moreoever, we weren’t able to fit our experimental data to simulations in the twisting regime; while twisting does lead to a decline in the buckling force, this decline is generally steeper than observed experimentally. Twisting in simulations eventually leads to an instability whereby a single segment accommodates all the twist (unphysical due to absence of steric interactions from simulation), causing a sudden sharp drop in the buckling force.

In addition, the twisting mode may not be consistent with the continued decline of buckling force observed experimentally; we expect that beyond a certain limit, twisting microbubules could become more rigid rather than soft, perhaps because of steric interactions, or breaking of bonds. In contrast, the cross-sectional ovalisation model allows for continued softening by the formation of increasingly more kinks. Further support for the ovalisation explanation comes from the fairly good agreement in shape between actual and simulated microtubules (see Figure 2).

We don’t know whether the microtubule can accommodate these high strains without undergoing structural damage. A lattice perturbation at the point of attachment between microtubule and optical bead may be expected to have immediate effect. In contrast, we see softening occur past some critical strain, whose value is consistent with the Brazier buckling mode, based on our fits.

- The mechanical lattice model is as simple as possible, which is a good thing since it limits the number of microscopic parameters. However, it is unclear whether a single shearing interaction (i.e. $\kappa_s$) in an effective 2D lattice is sufficient to stabilize the deformation of the lattice at the kinking instability. Furthermore, there seems to be some ambiguity concerning the relations between microscopic and macroscopic elastic constants (i.e. two different effective tube wall thicknesses are postulated, see. Table 2), which limits the quality of the predicted macroscopic elastic constants. This problem could be avoided by using for example a double shell lattice (with some diagonal elastic elements) which naturally includes stability against shear and bending in all directions.

A single shearing interaction fails, in fact, to stabilize the lattice deformation at the kinking instability unless the system is highly overdamped. The failure results in a complete cross-sectional collapse (in the absence of steric interactions) accompanied by a sharp, sudden drop in the buckling force and is likely due to the fact that in the 2D shell model there is no energy penalty for cross-sectional shear.

Since simulations under extremely damped conditions required to avoid complete cross-sectional collapse at the kinking instability are very time-consuming, we employ a shortcut in using slightly lower damping while fixing the positions of the middle (top and bottom) points of each cross-section in the transverse plane, preventing them from breaking transverse symmetry (thus preventing collapse through shearing), yet still allowing for cross-sectional ovalisation. Two different effective thicknesses are used due to the corrugated profile of microtubules. When modeled as a shell of uniform thickness, the actual cross section of a microtubule is replaced by an annular cross section with equivalent thickness *h* ≈ 2.7 nm (de Pablo et al., 2003). However, the effective bending stiffness is different; since most of the strain is localized to the bridges between the protofilaments (Schaap et al., 2006), the effective bending stiffness is closer to the bridge thickness. A double shell lattice (or a spoke-like geometry, perhaps) can certainly stabilize against cross-sectional shear, for example. However, this comes at a cost – increasing the number of parameters and the complexity of the model and possibly introducing reduncancy/degeneracy in the relations between microscopic and macroscopic elastic constants. For instance, circumferential stretching is now governed by two microscopic constants – the stiffnesses of both the diagonal and circumferential elements – such that the same macroscopic modulus may correspond to different combinations of the microscopic constants. Since we are able to use symmetry constraints to stabilize the much simpler single shell model, we may argue that a more complex model is not necessary.